# Deep Conditional Gaussian Mixture Model for Constrained Clustering

**Laura Manduchi**
Department of Computer Science
ETH Zürich
laura.manduchi@inf.ethz.ch

**Kieran Chin-Cheong**
Department of Computer Science
ETH Zürich
kieran.chincheong@inf.ethz.ch

**Holger Michel**
Department of Neonatology
University Children's Hospital Regensburg (KUNO)
University of Regensburg, Germany
holger.michel@barmherzige-regensburg.de

**Sven Wellmann**
Department of Neonatology
University Children's Hospital Regensburg (KUNO)
University of Regensburg, Germany
sven.wellmann@klinik.uni-regensburg.de

**Julia E. Vogt**
Department of Computer Science
ETH Zürich
julia.vogt@inf.ethz.ch

## Abstract

Constrained clustering has gained significant attention in the field of machine learning as it can leverage prior information on a growing amount of only partially labeled data. Following recent advances in deep generative models, we propose a novel framework for constrained clustering that is intuitive, interpretable, and can be trained efficiently in the framework of stochastic gradient variational inference. By explicitly integrating domain knowledge in the form of probabilistic relations, our proposed model (DC-GMM) uncovers the underlying distribution of data conditioned on prior clustering preferences, expressed as *pairwise constraints*. These constraints guide the clustering process towards a desirable partition of the data by indicating which samples should or should not belong to the same cluster. We provide extensive experiments to demonstrate that DC-GMM shows superior clustering performances and robustness compared to state-of-the-art deep constrained clustering methods on a wide range of data sets. We further demonstrate the usefulness of our approach on two challenging real-world applications.

## 1 Introduction

The ever-growing amount of data and the time cost associated with its labeling has made clustering a relevant task in the field of machine learning. Yet, in many cases, a fully unsupervised clustering algorithm might naturally find a solution that is not consistent with the domain knowledge (Basu et al., 2008). In medicine, for example, clustering could be driven by unwanted bias, such as the type of machine used to record the data, rather than more informative features. Moreover, practitioners often have access to prior information about the types of clusters that are sought, and a principled method to guide the algorithm towards a desirable configuration is then needed. Therefore, *constrained clustering* has a long history in machine learning as it enforces desirable clustering properties by

35th Conference on Neural Information Processing Systems (NeurIPS 2021).

incorporating domain knowledge, in the form of *instance-level* constraints (Wagstaff & Cardie, 2000), into the clustering objective.

A variety of methods have been proposed to extend deterministic deep clustering algorithms, such as DEC (Xie et al., 2016), to force the clustering process to be consistent with given constraints (Ren et al., 2019; Zhang et al., 2019). This results in a wide range of empirically motivated loss functions that are rather obscure in their underlying assumptions. Further, they are unable to uncover the distribution of the data, preventing them from being extended to other tasks beyond clustering, such as Bayesian model validation, outlier detection, and data generation (Min et al., 2018). Thus, we restrict our search for a constrained clustering approach to the class of deep generative models. Although these models have been successfully used in the unsupervised setting (Jiang et al., 2017; Dilokthanakul et al., 2016), their application to constrained clustering has been under-explored.

In this work we propose a novel probabilistic approach to constrained clustering, the Deep Conditional Gaussian Mixture Model (DC-GMM), that employs a deep generative model to uncover the underlying data distribution conditioned on domain knowledge, expressed in the form of pairwise constraints. Our model assumes a Conditional Mixture-of-Gaussians prior on the latent representation of the data. That is, a Gaussian Mixture Model conditioned on the user's clustering preferences, based e.g. on domain knowledge. These preferences are expressed as Bayesian prior probabilities with varying degrees of uncertainty. By integrating prior information in the generative process of the data, our model can guide the clustering process towards the configuration sought by the practitioners. Following recent advances in variational inference (Kingma & Welling, 2014; Rezende et al., 2014), we derive a scalable and efficient training scheme using amortized inference.

**Our main contributions** are as follows: (i) We propose a new paradigm for constrained clustering (DC-GMM) to incorporate instance-level clustering preferences, with varying degrees of certainty, within the Variational Auto-Encoder (VAE) framework. (ii) We provide a thorough empirical assessment of our model. In particular, we show that (a) a small fraction of prior information remarkably increases the performance of DC-GMM compared to unsupervised variational clustering methods, (b) our model shows superior clustering performance compared to state-of-the-art deep constrained clustering models on a wide range of data sets and, (c) our model proves to be robust against noise as it easily incorporate the uncertainty of the given constraints. (iii) Additionally, we demonstrate on two challenging real-world applications that our model can drive the clustering performance towards different desirable configurations, depending on the constraints used.

## 2 Deep Conditional Gaussian Mixture Model

In the following section, we propose a probabilistic approach to constrained clustering (DC-GMM) that incorporates clustering preferences, with varying degrees of certainty, in a VAE-based setting. In particular, we first describe the generative assumptions of the data conditioned on the domain knowledge, for which VaDE (Jiang et al., 2017) and GMM-VAE (Dilokthanakul et al., 2016) are special cases. We then define a concrete prior formulation to incorporate pairwise constraints and we derive a new objective, the Conditional ELBO, to train the model in the framework of stochastic gradient variational Bayes. Finally, we discuss the optimization procedure and the computational complexity of the proposed algorithm.

### 2.1 The Generative Assumptions

Let us consider a data set $\boldsymbol{X} = \{\boldsymbol{x}_i\}_{i=1}^{N}$ consisting of $N$ samples with $\boldsymbol{x}_i \in \mathbb{R}^M$ that we wish to cluster into $K$ groups according to instance-level prior information encoded as $\boldsymbol{W} \in \mathbb{R}^{N \times N}$. For example, we may know *a priori* that certain samples should (or should not) be clustered together. However, the prior information often comes from different sources with different noise levels. As an example, the instance-level annotations could be obtained from both very experienced domain experts and less experienced users. Hence, $\boldsymbol{W}$ should

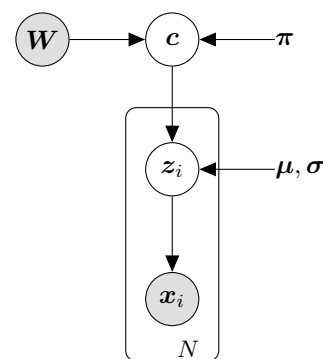

Figure 1: The proposed graphical model.

encode both our prior knowledge of the data set, expressed in the form of constraints, and its degree of confidence.

We assume the data is generated from a random process consisting of three steps, as depicted in Fig. 1. First, the cluster assignments $\mathbf{c} = \{c_i\}_{i=1}^N$, with $c_i \in \{1, \dots, K\}$, are sampled from a distribution conditioned on the prior information:

$$\mathbf{c} \sim p(\mathbf{c}|\boldsymbol{W}; \boldsymbol{\pi}) \tag{1}$$

The prior distribution of the cluster assignments without domain knowledge $\boldsymbol{W}$, i.e. $p(\mathbf{c})$, follows a categorical distribution with mixing parameters $\boldsymbol{\pi}$. Second, for each cluster assignment $c_i$, a continuous latent embedding, $\mathbf{z}_i \in \mathbb{R}^D$, is sampled from a Gaussian distribution, whose mean and variance depend on the selected cluster $c_i$. Finally, the sample $\mathbf{x}_i$ is generated from a distribution conditioned on $\mathbf{z}_i$. Given $c_i$, the generative process can be summarized as:

$$\mathbf{z}_i \sim p(\mathbf{z}_i|c_i) = \mathcal{N}(\mathbf{z}_i|\boldsymbol{\mu}_{c_i}, \boldsymbol{\sigma}_{c_i}^2 \mathbb{I}) \tag{2}$$

$$\mathbf{x}_i \sim p_\theta(\mathbf{x}_i|\mathbf{z}_i) = \begin{cases} \mathcal{N}(\mathbf{x}_i|\boldsymbol{\mu}_{x_i}, \boldsymbol{\sigma}_{x_i}^2 \mathbb{I}) & \text{if real-valued} \\ Ber(\boldsymbol{\mu}_{x_i}) & \text{if binary} \end{cases} \tag{3}$$

where $\boldsymbol{\mu}_{c_i}$ and $\boldsymbol{\sigma}_{c_i}^2$ are mean and variance of the Gaussian distribution corresponding to cluster $c_i$ in the latent space. In the case where x is real-valued then $[\boldsymbol{\mu}_{x_i}, \boldsymbol{\sigma}_{x_i}^2] = f(\mathbf{z}_i; \boldsymbol{\theta})$, if x is binary then $\boldsymbol{\mu}_{x_i} = f(\mathbf{z}_i; \boldsymbol{\theta})$. The function $f(\mathbf{z}; \boldsymbol{\theta})$ denotes a neural network, called *decoder*, parametrized by $\boldsymbol{\theta}$.

It is worth noting that, given $\boldsymbol{W}$, the cluster assignments are not necessarily independent, i.e. there might be certain $i, j \in \{1, \dots, N\}$ for which $(c_i \not\perp\!\!\!\perp c_j | \boldsymbol{W})$. This important detail prevents the use of standard optimization procedure and it will be explored in the following Sections. On the contrary, if there is no prior information, that is $p(\mathbf{c}|\boldsymbol{W}) = p(\mathbf{c}) = \prod_i p(c_i) = \prod_i Cat(c_i|\boldsymbol{\pi})$, the cluster assignments are independent and identical distributed. In that particular case, the generative assumptions described above are equal to those of Jiang et al. (2017); Dilokthanakul et al. (2016) and the parameters of the model can be learned using the unsupervised VaDE method. As a result, VaDE (or GMM-VAE) can be seen as a special case of our framework.

## 2.2 Conditional Prior Probability with Pairwise Constraints

We incorporate the clustering preference through the conditional probability $p(\mathbf{c}|\boldsymbol{W})$. We focus on pairwise constrains consisting of *must-links*, if two samples are believed to belong to the same cluster, and *cannot-links*, otherwise. However, different types of constraints can be included, for example, triple-constraints (see Appendix C).

**Definition 1.** *Given a dataset $\boldsymbol{X} = \{\boldsymbol{x}_i\}_{i=1}^N$, the pairwise prior information $\boldsymbol{W} \in \mathbb{R}^{N \times N}$ is defined as a symmetric matrix containing the pairwise preferences and confidence. In particular*

$$\boldsymbol{W}_{i,j} \begin{cases} > 0 & \text{if there is a must-link constraint between } \boldsymbol{x}_i \text{ and } \boldsymbol{x}_j \\ = 0 & \text{if there is no prior information on samples } \boldsymbol{x}_i \text{ and } \boldsymbol{x}_j \\ < 0 & \text{if there is a cannot-link constraint between } \boldsymbol{x}_i \text{ and } \boldsymbol{x}_j, \end{cases}$$

*where the value $|\boldsymbol{W}_{i,j}| \in [0, \infty)$ reflects the degree of certainty in the constraint.*

**Definition 2.** *Given the pairwise prior information $\boldsymbol{W}$, the conditional prior probability $p(\mathbf{c}|\boldsymbol{W}; \boldsymbol{\pi})$ is defined as:*

$$p(\mathbf{c}|\boldsymbol{W}; \boldsymbol{\pi}) := \frac{\prod_i \pi_{c_i} h_i(\mathbf{c}, \boldsymbol{W})}{\sum_{\mathbf{c}} \prod_j \pi_{c_j} h_j(\mathbf{c}, \boldsymbol{W})} = \frac{1}{\Omega(\boldsymbol{\pi})} \prod_i \pi_{c_i} h_i(\mathbf{c}, \boldsymbol{W}), \tag{4}$$

*where $\boldsymbol{\pi} = \{\pi_k\}_{k=1}^K$ are the weights associated to each cluster, $c_i$ is the cluster assignment of sample $\mathbf{x}_i$, $\Omega(\boldsymbol{\pi})$ is the normalization factor and $h_i(\mathbf{c}, \boldsymbol{W})$ is a weighting function of the form*

$$h_i(\mathbf{c}, \boldsymbol{W}) = \prod_{j \neq i} \exp\left(\boldsymbol{W}_{i,j} \delta_{c_i c_j}\right). \tag{5}$$

It follows that $h_i(\mathbf{c}, \boldsymbol{W})$ assumes large values if $c_i$ agrees with our belief with respect to $\mathbf{c}$ and low values otherwise. If $\boldsymbol{W}_{i,j} \to -\infty$ then $\boldsymbol{x}_i$ and $\boldsymbol{x}_j$ must be assigned to different clusters otherwise

$p(\mathbf{c}|\boldsymbol{W}) \to 0$ (*hard* constraint). On the other hand, smaller values indicate a *soft preference* as they admit some degree of freedom in the model. An heuristic to select $|\boldsymbol{W}_{i,j}|$ is presented in Sec 4.

The conditional prior probability with pairwise constraints has been successfully used by traditional clustering methods in the past (Lu & Leen, 2004), but to the best of our knowledge, it has never been applied in the context of deep generative models. It can also be seen as the posterior of the superparamagnetic clustering method (Blatt et al., 1996), with loss function given by a fully connected Potts model (Wu, 1982).

## 2.3 Conditional Evidence Lower Bound

Given the data generative assumptions illustrated in Sec. 2.1, the objective is to infer the parameters of the model, $\boldsymbol{\theta}$, $\boldsymbol{\pi}$, and $\boldsymbol{\nu} = \{\boldsymbol{\mu}_c, \boldsymbol{\sigma}_c^2\}_{c=1}^{K}$, given both the observed data $\mathbf{X}$ and the pairwise prior information on the cluster assignments $\boldsymbol{W}$. This could be achieved by maximizing the marginal log-likelihood conditioned on $\boldsymbol{W}$, that is:

$$\log p(\mathbf{X}|\boldsymbol{W}) = \log \int_{\mathbf{Z}} \sum_{\mathbf{c}} p(\mathbf{X}, \mathbf{Z}, \mathbf{c}|\boldsymbol{W}), \tag{6}$$

where $\mathbf{Z} = \{\boldsymbol{z}_i\}_{i=1}^{N}$ is the collection of the latent embeddings corresponding to the data set $\boldsymbol{X}$. The conditional joint probability is derived from Eq. 2 and Eq. 3 and can be factorized as:

$$p(\mathbf{X}, \mathbf{Z}, \mathbf{c}|\boldsymbol{W}) = p_{\boldsymbol{\theta}}(\mathbf{X}|\mathbf{Z})p(\mathbf{Z}|\mathbf{c}; \boldsymbol{\nu})p(\mathbf{c}|\boldsymbol{W}; \boldsymbol{\pi}) = p(\mathbf{c}|\boldsymbol{W}; \boldsymbol{\pi}) \prod_{i=1}^{N} p_{\boldsymbol{\theta}}(\mathbf{x}_i|\mathbf{z}_i)p(\mathbf{z}_i|\mathbf{c}_i; \boldsymbol{\nu}). \tag{7}$$

Since the conditional log-likelihood is intractable, we derive an alternative tractable objective.

**Definition 3.** $\mathcal{L}_{\mathrm{C}}$, *the Conditional ELBO (C-ELBO), is defined as*

$$\mathcal{L}_{\mathrm{C}}(\boldsymbol{\theta}, \boldsymbol{\phi}, \boldsymbol{\nu}, \boldsymbol{\pi}, \mathbf{X}|\boldsymbol{W}) := \mathbb{E}_{q_{\boldsymbol{\phi}}(\mathbf{Z}|\mathbf{X})}\left[\log p_{\boldsymbol{\theta}}(\mathbf{X}|\mathbf{Z})\right] - D_{KL}(q_{\boldsymbol{\phi}}(\mathbf{Z}, \mathbf{c}|\mathbf{X}) \| p(\mathbf{Z}, \mathbf{c}|\boldsymbol{W}; \boldsymbol{\nu}, \boldsymbol{\pi})), \tag{8}$$

*with* $q_{\boldsymbol{\phi}}(\mathbf{Z}, \mathbf{c}|\mathbf{X})$ *being the following amortized mean-field variational distribution:*

$$q_{\boldsymbol{\phi}}(\mathbf{Z}, \mathbf{c}|\mathbf{X}) = q_{\boldsymbol{\phi}}(\mathbf{Z}|\mathbf{X})p(\mathbf{c}|\mathbf{Z}; \boldsymbol{\nu}, \boldsymbol{\pi}) = \prod_{i=1}^{N} q_{\boldsymbol{\phi}}(\mathbf{z}_i|\mathbf{x}_i)p(\mathbf{c}_i|\mathbf{z}_i; \boldsymbol{\nu}, \boldsymbol{\pi}). \tag{9}$$

The first term of the Conditional ELBO is known as the *reconstruction term*, similarly to the VAE. The second term, on the other hand, is the Kullback-Leibler (KL) divergence between the variational posterior and the Conditional Gaussian Mixture prior. By maximizing the C-ELBO, the variational posterior mimics the true conditional probability of the latent embeddings and the cluster assignments. This results in enforcing the latent embeddings to follow a Gaussian mixture that agrees on the clustering preferences.

From Definition 3 and Equations 6 and 7, we can directly derive Lemma 1:

**Lemma 1.** *It holds that*

1. *The C-ELBO* $\mathcal{L}_{\mathrm{C}}$ *is a lower bound of the marginal log-likelihood conditioned on* $\boldsymbol{W}$, *that is*

$$\log p(\mathbf{X}|\boldsymbol{W}) \geq \mathcal{L}_{\mathrm{C}}(\boldsymbol{\theta}, \boldsymbol{\phi}, \boldsymbol{\nu}, \boldsymbol{\pi}, \mathbf{X}|\boldsymbol{W}).$$

2. $\log p(\mathbf{X}|\boldsymbol{W}) = \mathcal{L}_{\mathrm{C}}(\boldsymbol{\theta}, \boldsymbol{\phi}, \boldsymbol{\nu}, \boldsymbol{\pi}, \mathbf{X}|\boldsymbol{W})$ *if and only if* $q_{\boldsymbol{\phi}}(\mathbf{Z}, \mathbf{c}|\mathbf{X}) = p(\mathbf{Z}, \mathbf{c}|\mathbf{X}, \boldsymbol{W})$.

For the proof we refer to the Appendix B. It is worth noting that in Eq. 9, the variational distribution does not depend on $\boldsymbol{W}$. This approximation is used to retain a mean-field variational distribution when the cluster assignments, conditioned on the prior information, are not independent (Sec 2.1), that is when $p(\mathbf{c}|\boldsymbol{W}) \neq \prod_i p(\mathbf{c}_i|\boldsymbol{W})$. Additionally, the probability $p(\mathbf{c}_i|\mathbf{z}_i)$ can be easily computed using the Bayes Theorem, yielding

$$p(\mathbf{c}_i|\mathbf{z}_i; \boldsymbol{\nu}, \boldsymbol{\pi}) = \frac{\mathcal{N}(\mathbf{z}_i|\boldsymbol{\mu}_{c_i}, \boldsymbol{\sigma}_{c_i}^2)\pi_{c_i}}{\sum_k \mathcal{N}(\mathbf{z}_i|\boldsymbol{\mu}_k, \boldsymbol{\sigma}_k^2)\pi_k}, \tag{10}$$

while we define the variational distribution $q_{\boldsymbol{\phi}}(\mathbf{z}_i|\mathbf{x}_i)$ to be a Gaussian distribution with mean $\boldsymbol{\mu}_{\boldsymbol{\phi}}(\boldsymbol{x}_i)$ and variance $\boldsymbol{\sigma}_{\boldsymbol{\phi}}^2(\boldsymbol{x}_i)\mathbb{I}$ parametrized by a neural network, also known as *encoder*.

## 2.4 Optimisation & Computational Complexity

The parameters of the generative model and the parameters of the variational distribution are optimised by maximising the C-ELBO. From Definitions 2 and 3, we derive Lemma 2:

**Lemma 2.** *The Conditional ELBO $\mathcal{L}_{\mathrm{C}}$ factorizes as follow:*

$$
\mathcal{L}_{\mathrm{C}}(\boldsymbol{\theta}, \boldsymbol{\phi}, \boldsymbol{\nu}, \boldsymbol{\pi}, \mathbf{X}|\boldsymbol{W}) = -\log \Omega(\boldsymbol{\pi}) + \sum_{i=1}^{N} E_{q_{\phi}(\mathbf{z}_i|\mathbf{x}_i)} \Big[ \log p_{\boldsymbol{\theta}}(\mathbf{x}_i|\mathbf{z}_i) - \log q_{\phi}(\mathbf{z}_i|\mathbf{x}_i) \Big]
$$

$$
+ \sum_{i=1}^{N} E_{q_{\phi}(\mathbf{z}_i|\mathbf{x}_i)} \sum_{k=1}^{K} p(k|\mathbf{z}_i) \Big[ \log p(\mathbf{z}_i|k) + \log \pi_{\mathrm{k}} - \log p(k|\mathbf{z}_i) \Big] \quad (11)
$$

$$
+ \sum_{i \neq j=1}^{N} E_{q_{\phi}(\mathbf{z}_i|\mathbf{x}_i)} E_{q_{\phi}(\mathbf{z}_j|\mathbf{x}_j)} \sum_{k=1}^{K} p(k|\mathbf{z}_i) p(k|\mathbf{z}_j) \boldsymbol{W}_{i,j},
$$

*where $p(k|\mathbf{z}_i) = p(\mathrm{c}_i = k|\mathbf{z}_i; \boldsymbol{\nu}, \boldsymbol{\pi})$ and $p(\mathbf{z}_i|k) = p(\mathbf{z}_i|c_i = k; \boldsymbol{\nu})$.*

For the complete proof we refer to the Appendix B. Maximizing Eq. 11 w.r.t. $\boldsymbol{\pi}$ poses computational problems due to the normalization factor $\Omega(\boldsymbol{\pi})$. Crude approximations are investigated in (Basu et al., 2008), however we choose to fix the parameter $\pi_k = 1/K$ to make $\mathbf{z}$ uniformly distributed in the latent space, as in previous works (Dilokthanakul et al., 2016). Hence the normalization factor can be treated as a constant. The Conditional ELBO can then be approximated using the SGVB estimator and the reparameterization trick (Kingma & Welling, 2014) to be trained efficiently using stochastic gradient descent. We refer to the Appendix B for the full derivation. We observe that the pairwise prior information only affects the last term, which scans through the dataset twice. To allow for fast iteration we simplify it by allowing the search of pairwise constraints to be performed only inside the considered batch, yielding

$$
\frac{1}{L} \sum_{l=1}^{L} \sum_{i \neq j=1}^{B} \sum_{k=1}^{K} p(\mathrm{c}_i = k|\mathbf{z}_i^{(l)}) p(\mathrm{c}_j = k|\mathbf{z}_j^{(l)}) \boldsymbol{W}_{i,j}, \quad (12)
$$

where $L$ denotes the number of Monte Carlo samples and $B$ the batch size. By doing so, the overhead in computational complexity of a single joint update of the parameters is $O(LB^2 K C_p^2)$, where $C_p$ is the cost of evaluating $p(\mathrm{c}_i = k|\mathbf{z}_i)$ with $\mathbf{z}_i \in \mathbb{R}^D$. The latter is $O(KD)$.

## 3 Related Work

**Constrained Clustering.** A constrained clustering problem differs from the classical clustering scenario as the user has access to some pre-existing knowledge about the desired partition of the data expressed as instance-level constraints (Lange et al., 2005). Traditional clustering methods, such as the well-known K-means algorithm, have been extended to enforce pairwise constraints (Wagstaff et al. (2001), Bilenko et al. (2004)). Several methods also proposed a constrained version of the Gaussian Mixture Models (Shental et al., 2003; Law et al., 2004, 2005). Among them, penalized probabilistic clustering (PPC, Lu & Leen (2004)) is the most related to our work as it expresses the pairwise constraints as Bayesian priors over the assignment of data points to clusters, similarly to our model. However, all previous mentioned models shows poor performance and high computational complexity on high-dimensional and large-scale data sets.

**Constrained Deep Clustering.** To overcome the limitations of the above models, constrained clustering algorithms have lately been used in combination with deep neural networks (DNNs). Hsu & Kira (2015) train a DNN to minimize the Kullback-Leibler (KL) divergence between similar pairs of samples, while Chen (2015) performs semi-supervised maximum margin clustering on the learned features of a DNN. More recently, many extensions of the widely used DEC model (Xie et al., 2016) have been proposed to include a variety of loss functions to enforce pairwise constraints. Among them, SDEC, (Ren et al., 2019) includes a distance loss function that forces the data points with a must-link to be close in the latent space and vice-versa. Constrained IDEC (Zhang et al., 2019), uses a KL divergence loss instead, extending the work of Shukla et al. (2018). Smieja et al. (2020) focuses on discriminative clustering methods by self-generating pairwise constraints from Siamese networks.

As none of these approaches are based on generative models, the above methods fail to uncover the underlying data distribution.

**Deep Generative Models.** Although a wide variety of generative models have been proposed in the literature to perform unsupervised clustering (Li et al., 2019; Yang et al., 2019; Manduchi et al., 2021; Jiang et al., 2017), not much effort has been directed towards extending them to incorporate domain knowledge and clustering preferences. Nevertheless, the inclusion of prior information on a growing amount of unlabelled data is of profound practical importance in a wide range of applications (Kingma et al., 2014). The only exception is the work of Luo et al. (2018), where the authors proposed the SDCD algorithm, which has remarkably lower clustering performance compared to state-of-the-art constrained clustering models, as we will show in the experiments. Different from our approach, SDCD models the joint distribution of data and pairwise constraints. The authors adopts the two-coin Dawid-Skene model from Raykar et al. (2010) to model $p(\boldsymbol{W}|\boldsymbol{c})$, resulting in a different graphical model (see Fig. 1). Instead, we consider a simpler and more intuitive scenario, where we assume the cluster assignments are conditioned on the prior information, $p(\mathbf{c}|\boldsymbol{W};\boldsymbol{\pi})$ (see Definition 2). In other words, we assume that different clustering structures might be present within a data set and the domain knowledge should indicate which one is preferred over the other. It is also worth mentioning the work of Tang et al. (2019), where the authors propose to augment the prior distribution of the VAE to take into account the correlations between samples. Contrarily to the DC-GMM, their proposed approach requires additional lower bounds as the ELBO cannot be directly applied. Additionally, it also requires post-hoc clustering of the learnt latent space to compute the cluster assignments.

# 4    Experiments

In the following, we provide a thorough empirical assessment of our proposed method (DC-GMM) with pairwise constraints using a wide range of data sets. First, we evaluate our model's performance compared to both the unsupervised variational deep clustering method and state-of-the-art constrained clustering methods. As a next step, we present extensive evidence of the ability of our model to handle noisy constraint information. Additionally, we perform experiments on a challenging real medical data set consisting of pediatric heart ultrasound videos, as well as a face image data set to demonstrate that our model can reach different desirable partitions of the data, depending on the constraints used, with real-world, noisy data.

**Baselines & Implementation Details.**    As baselines, we include the traditional pairwise constrained K-means (PCKmeans, Basu et al. (2004)) and two recent deterministic deep constrained clustering methods based on DEC (SDEC, Ren et al. (2019), and Constrained IDEC, Zhang et al. (2019)) as they achieve state-of-the-art performance in constrained clustering. For simplicity, we will refer to the latter as C-IDEC. We also compare our method to generative models, the semi-supervised SCDC (Luo et al., 2018) and the unsupervised VaDE (Jiang et al., 2017). To implement our model, we were careful in maintaining a fair comparison with the baselines. In particular, we adopted the same encoder and decoder feed-forward architecture used by the baselines: four layers of 500, 500, 2000, $D$ units respectively, where $D = 10$ unless stated otherwise. The VAE is pretrained for 10 epochs while the DEC-based baselines need a more complex layer-wise pretraining of the autoencoder which involves 50 epochs of pretraining for each layer and 100 epochs of pretraining as finetuning. Each data set is divided into training and test sets, and all the reported results are computed on the latter. We employed the same hyper-parameters for all data sets, see Appendix F.1 for details. The pairwise constraints are chosen randomly within the training set by sampling two data points and assigning a must-link if they have the same label and a cannot-link otherwise. Unless stated otherwise, the values of $|W_{i,j}|$ are set to $10^4$ for all data sets, and 6000 pairwise constraints are used for both our model and the constrained clustering baselines. Note that the total amount of pairwise annotations in a data set of length $N$ is $O(N^2)$. Given that $N$ is typically larger than 10000, the number of pairwise constraints used in the experiments represents a small fraction of the total information.

**Constrained clustering.**    We first compare the clustering performance of our model with the baselines on four different standard data sets: MNIST (LeCun et al., 2010), Fashion MNIST (Xiao et al., 2017), Reuters (Xie et al., 2016) and STL-10 (Coates et al., 2011) (see Appendix A). More complex data sets will be explored in the following paragraphs. Note that we pre-processed the Reuters data by computing the tf-idf features on the 2000 most frequent words on a random subset of 10 000 documents and by selecting 4 root categories (Xie et al., 2016). Additionally, we extracted

Table 1: Clustering performances (%) of our proposed method DC-GMM compared with baselines. All methods use 6000 pairwise constraints except the unsupervised VaDE and the SCDC. Means and standard deviations are computed across 10 runs with different random model initialization. *VaDE results are different from (Jiang et al., 2017) as they only report their best performance. **Results taken from (Luo et al., 2018).

| Dataset | Metric | VaDE* | PCKmeans | SDEC | C-IDEC | SCDC** | DC-GMM (ours) |
|---------|--------|-------|----------|------|--------|--------|---------------|
| MNIST   | Acc | $89.0_{\pm5.0}$ | $56.4_{\pm2.0}$ | $86.2_{\pm0.1}$ | $96.3_{\pm0.2}$ | 84.2 | $\mathbf{96.6}_{\pm0.1}$ |
|         | NMI | $82.8_{\pm3.0}$ | $50.5_{\pm1.3}$ | $84.2_{\pm0.1}$ | $\mathbf{91.8}_{\pm1.0}$ | 81.2 | $91.5_{\pm0.2}$ |
|         | ARI | $80.9_{\pm5.0}$ | $38.8_{\pm1.9}$ | $80.1_{\pm0.1}$ | $92.1_{\pm0.4}$ | - | $\mathbf{92.7}_{\pm0.3}$ |
| FASHION | Acc | $55.1_{\pm2.2}$ | $53.9_{\pm2.9}$ | $54.0_{\pm0.2}$ | $68.1_{\pm3.0}$ | - | $\mathbf{80.0}_{\pm1.0}$ |
|         | NMI | $57.9_{\pm2.7}$ | $50.8_{\pm1.3}$ | $57.3_{\pm0.1}$ | $66.7_{\pm2.0}$ | - | $\mathbf{71.8}_{\pm0.5}$ |
|         | ARI | $41.6_{\pm3.1}$ | $36.1_{\pm1.7}$ | $40.2_{\pm0.1}$ | $52.3_{\pm3.0}$ | - | $\mathbf{65.8}_{\pm0.7}$ |
| REUTERS | Acc | $76.0_{\pm0.7}$ | $71.5_{\pm2.4}$ | $82.1_{\pm0.1}$ | $94.7_{\pm0.6}$ | - | $\mathbf{95.4}_{\pm0.2}$ |
|         | NMI | $50.1_{\pm1.3}$ | $48.2_{\pm3.8}$ | $62.3_{\pm0.1}$ | $81.4_{\pm0.7}$ | - | $\mathbf{82.7}_{\pm0.7}$ |
|         | ARI | $58.0_{\pm1.4}$ | $46.5_{\pm4.2}$ | $66.7_{\pm0.1}$ | $87.7_{\pm0.9}$ | - | $\mathbf{89.0}_{\pm0.6}$ |
| STL-10  | Acc | $77.3_{\pm0.5}$ | $70.3_{\pm4.2}$ | $79.2_{\pm0.1}$ | $81.6_{\pm3.8}$ | - | $\mathbf{89.5}_{\pm0.5}$ |
|         | NMI | $70.6_{\pm0.4}$ | $71.6_{\pm1.3}$ | $78.6_{\pm0.1}$ | $77.3_{\pm1.7}$ | - | $\mathbf{80.2}_{\pm0.7}$ |
|         | ARI | $62.7_{\pm0.4}$ | $58.4_{\pm2.1}$ | $71.0_{\pm0.1}$ | $71.8_{\pm3.4}$ | - | $\mathbf{78.4}_{\pm0.9}$ |

features from the STL-10 image data set using a ResNet-50 (He et al., 2016), as in previous works (Jiang et al., 2017). Accuracy, Normalized Mutual Information (NMI), and Adjusted Rand Index (ARI) are used as evaluation metrics. In Table 1 we report the mean and standard deviation of the clustering performance across 10 runs of both our method and the baselines. The only exception is the SDCD, for which we only report their original results (Luo et al., 2018) computed with a higher number of constraints. The provided code with 6000 constraints produced highly unstable and sub-optimal results (see Appendix D).

We observe that our model reaches state-of-the-art clustering performance in almost all metrics and data sets. As C-IDEC turns out to be the strongest baseline, we performed additional comparison to investigate the difference in performance under different settings. In Figure 2 we plot the clustering performance in terms of ARI using a varying number of constraints, $N_c$. For additional metrics, we refer to the Appendix E.1. We observe that our method outperforms the strongest baseline C-IDEC by a large margin on all four data sets when fewer constraints are used. When $N_c$ gets close to $N$, the two methods tend to saturate on the Reuters and STL data sets.

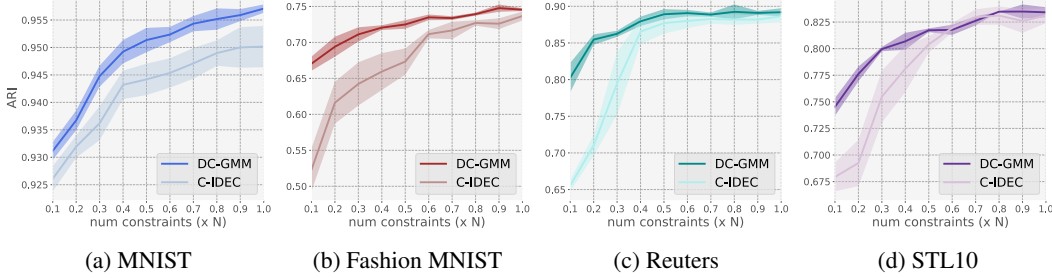

|   (a) MNIST   |   (b) Fashion MNIST   |   (c) Reuters   |   (d) STL10   |

Figure 2: Comparison of clustering performance of our proposed DC-GMM model and the strongest baseline C-IDEC on four different data sets. The number of constraints vary between $0.1 \times N$ and $N$, where $N$ is the length of the data set. ARI is used as evaluation metric.

**Constrained clustering with noisy labels.** In real-world applications it is often the case that the additional information comes from different sources with different confidence levels. Hence, the ability to integrate constraints with different degrees of certainty into the clustering algorithm is of significant practical importance. In this experiment, we consider the case in which the given pairwise constraints have three different noise levels, $q \in \{0.1, 0.2, 0.3\}$, where $q$ determines the fraction

of pairwise constraints with flipped signs (that is, when a must-link is turned into a cannot-link and vice-versa). In Fig. 3 we show the ARI clustering performance of our model compared to the strongest baseline derived from the previous section, C-IDEC. For all data sets, we decrease the value of the pairwise confidence of our method using the heuristic $|W_{i,j}| = \alpha \log\left(\frac{1-q}{q}\right)$ with $\alpha = 1000$. Also, we use grid search to choose the hyper-parameters of C-IDEC for the different noise levels (in particular we set the penalty weight of their loss function to 0.01, 0.005, and 0.001 respectively). Additionally, we report Accuracy and NMI in Appendix E.2. DC-GMM clearly achieves better performance on all three noise levels for all data sets. In particular, the higher the noise level, the greater the difference in performance. We conclude that our model is more robust than its main competitor on noisy labels and it can easily include different sources of information with different degrees of uncertainty.

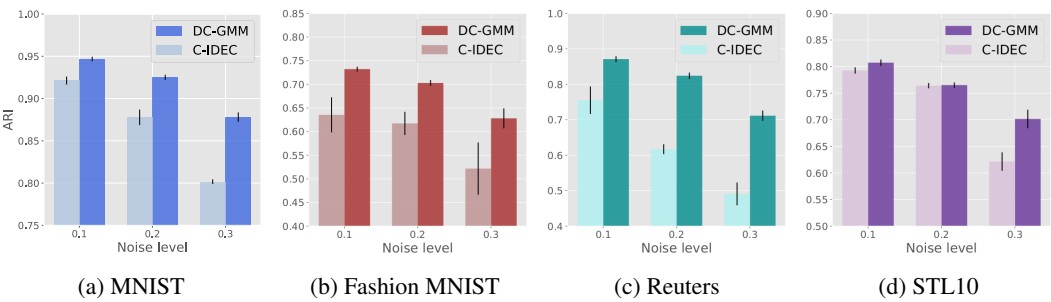

|        (a) MNIST        |        (b) Fashion MNIST        |        (c) Reuters        |        (d) STL10        |

Figure 3: Comparison of clustering performance of our proposed DC-GMM model and the strongest baseline C-IDEC on four different data sets with noisy labels. ARI is used as evaluation metric.

**Heart Echo.** We evaluate the capability of our model in a real-world application by using a data set consisting of 305 infant echo cardiogram videos obtained from the Hospital Barmherzige Brüder Regensburg. The videos are taken from five different angles (called views), denoted by [LA, KAKL, KAPAP, KAAP, 4CV]. Preprocessing of the data includes cropping the videos, resizing them to $64 \times 64$ pixels and splitting them into a total of 20000 individual frames. We investigated two different constrained clustering settings. First, we cluster the echo video frames by view (Zhang et al., 2018). Then, we cluster the echo video frames by infant maturity at birth, following the WHO definition of premature birth categories ("Preterm"). We believe that these two clustering tasks demonstrate that our model admits a degree of control in choosing the underlying structure of the learned clusters.
For both experiments, we compare the performance of our method with the unsupervised VaDE and with C-IDEC. Additionally, we include a variant of both our method and VaDE in which we use a VGG-like convolutional neural network (Simonyan & Zisserman, 2015) , for details on the implementation we refer to the Appendix F.2. The results are shown in Table 2. The DC-GMM outperforms both baselines by a significant margin in accuracy, NMI, and ARI, as well as in both clustering experiments. We also observe that C-IDEC performs poorly on real-world noisy data. We believe this is due to the heavy pretraining of the autoencoder, required by DEC-based methods, as it

Table 2: Clustering performance (%) using the heart echo cardiogram data with fully connected layers on the left and convolutional layers (CNN-) on the right. All methods use 6000 pairwise constraints except the VaDE. Means and standard deviations are computed across 10 runs with different random model initialization.

| Clustering | Metric | VaDE | C-IDEC | **DC-GMM** | CNN-VaDE | **CNN-DC-GMM** |
|---|---|---|---|---|---|---|
| View | Acc | 33.4 ±3.3 | 55.1 ±16.0 | **83.2** ±1.4 | 41.7 ±5.2 | **92.5** ±1.4 |
|  | NMI | 8.9 ±2.7 | 33.3 ±15.9 | **64.9** ±2.3 | 19.7 ±7.4 | **82.6** ±3.0 |
|  | ARI | 6.5 ±2.5 | 31.2 ±15.1 | **63.7** ±2.7 | 13.7 ±6.3 | **83.1** ±3.1 |
| Preterm | Acc | 41.4 ±5.4 | 69.6 ±1.1 | **72.3** ±1.5 | 36.7 ±2.5 | **73.3** ±1.1 |
|  | NMI | 6.4 ±1.8 | 8.3 ±11.8 | **25.1** ±3.4 | 6.5 ±4.0 | **32.0** ±2.4 |
|  | ARI | 2.3 ±2.6 | 13.7 ±19.3 | **45.1** ±4.1 | 3.9 ±3.8 | **48.1** ±2.8 |

does not always lead to a learned latent space that is suitable for the clustering task. Additionally, we illustrate a PCA decomposition of the embedded space learned by both the DC-GMM and the unsupervised VaDE baseline for both tasks in Figure 4. Our method is clearly able to learn an embedded space that clusters both the different views and the different preterms more effectively than the unsupervised VaDE. This observation, together with the quantitative results, demonstrates that adding domain knowledge is particularly effective for medical purposes.

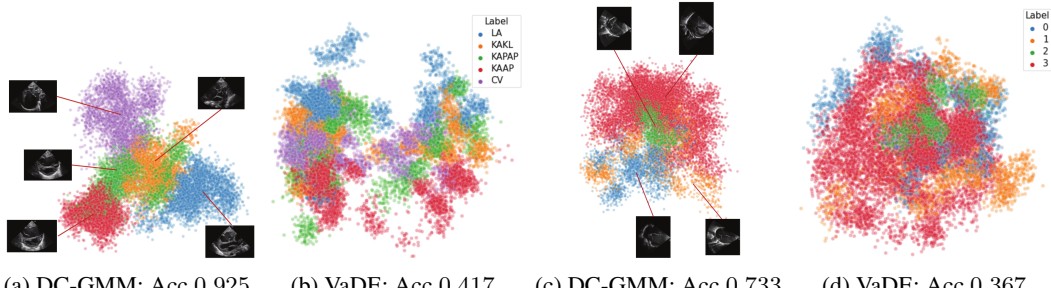

(a) DC-GMM: Acc 0.925    (b) VaDE: Acc 0.417    (c) DC-GMM: Acc 0.733    (d) VaDE: Acc 0.367

Figure 4: PCA decomposition of heart ultrasound imaging test set examples in the embedded space and generative samples using the unsupervised VaDE and our model DC-GMM with 6000 constraints for (a)-(b) View, (c)-(d) Preterm.

**Face Images.** We further evaluate the performance of our model using the UTKFace data set (Zhang et al., 2017). This data set contains over 20000 images of male and female faces, aged from 1 to 118 years old, with multiple ethnicities represented. We use VGG nets (Simonyan & Zisserman, 2015) for the VAE (the implementation details are described in the Appendix F.3). As in the Heart Echo experiment, we chose two different clustering tasks. First we cluster the data using the gender prior information, then we select a sub-sample of individuals between 18 and 50 years of age (approx. 11000 samples) and cluster by ethnicity (White, Black, Indian, Asian). In Fig. 5 we illustrate the PCA decomposition of the embedded space learned by the VaDE and our model. For both tasks we use $2N$ pairwise constraints where $N$ is the length of the data set, which requires labels for $1.5\%$ of the entire data set. Specifically, on the gender task and the ethnicity task our model achieves an accuracy of $0.89$ and $0.85$, which outperforms VaDE with a relative increase ratio of $74.5\%$ and $49.1\%$. In terms of NMI, the unsupervised VaDE performance is close to 0 in both tasks, while our model performance is $0.52$ and $0.54$ respectively. Visually, we observe a neat division of the selected clusters in the embedding space with the inclusion of domain knowledge. The unsupervised approach is not able to distinguish any feature of interest. We conclude that it is indeed possible to guide the clustering process towards a preferred configuration, depending on what the practitioners are seeking in the data, by providing different pairwise constraints. Finally, we tested the generative capabilities of our model by sampling from the learnt generative process of Sec 2.1. For a visualization of the generated sample we refer to the Appendix E.3.

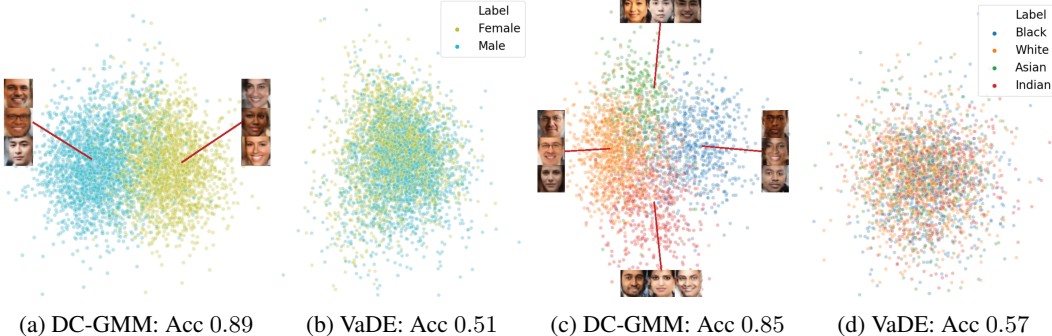

(a) DC-GMM: Acc 0.89    (b) VaDE: Acc 0.51    (c) DC-GMM: Acc 0.85    (d) VaDE: Acc 0.57

Figure 5: PCA decomposition of test set examples in the embedded space and generative samples using VaDE and DC-GMM for (a)-(b) Gender, (c)-(d) Ethnicity. In this configuration, DC-GMM obtains a NMI of $0.52$ (gender) and $0.58$ (ethnicity) while VaDE obtains a NMI close to 0 for both tasks.

# 5 Conclusion

In this work, we present a novel constrained deep clustering method called DC-GMM, that incorporates clustering preferences in the form of pairwise constraints, with varying degrees of certainty. In contrast to existing deep clustering approaches, DC-GMM uncovers the underlying distribution of the data conditioned on prior clustering preferences. With the integration of domain knowledge, we show that our model can drive the clustering algorithm towards the partitions of the data sought by the practitioners, achieving state-of-the-art constrained clustering performance in real-world and complex data sets. Additionally, our model proves to be robust to noisy constraints as it can efficiently include uncertainty into the clustering preferences. As a result, the proposed model can be applied to a variety of applications where the difficulty of obtaining labeled data prevents the use of fully supervised algorithms.

**Limitations & Future Work**   The proposed algorithm requires that the mixing parameters $\pi$ of the clusters are chosen *a priori*. This limitation is mitigated by the prior information $W$, which permits a more flexible prior distribution if enough information is available (see Definition 2). The analysis of different approaches to learn the weights $\pi$ represents a potential direction for future work. Additionally, the proposed framework could also be used in a self-supervised manner, by learning $W$ from the data using, e.g. contrastive learning approaches (Chen et al., 2020; Wu et al., 2018).

# 6 Code and Data Availability

The code is available in a GitHub repository: https://github.com/lauramanduchi/DC-GMM. All datasets are publicly available except the Heart Echo data. The latter is not available due to medical confidentiality.

## Acknowledgments and Disclosure of Funding

We would like to thank Luca Corinzia (ETH Zurich) for the fruitful discussions that contributed to shape this work. We would also like to thank Vincent Fortuin (ETH Zurich), Thomas M. Sutter (ETH Zurich), Imant Daunhawer (ETH Zurich), and Ričards Marcinkevičs (ETH Zurich) for their helpful comments and suggestions. This project has received funding from the PHRT SHFN grant #1-000018-057: SWISSHEART.

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
