# Appendix

## A   Data sets

The data sets used in the experiments are the followings:

- **MNIST:** It consists of $70\,000$ handwritten digits. The images are centered and of size 28 by 28 pixels. We reshaped each image to a 784-dimensional vector (LeCun et al., 2010).
- **Fashion MNIST:** A data set of Zalando's article images consisting of a training set of $60\,000$ examples and a test set of $10\,000$ examples (Xiao et al., 2017).
- **Reuters:** It contains $810\,000$ English news stories (Lewis et al., 2004). Following the work of Xie et al. (2016), we used 4 root categories: corporate/industrial, government/social, markets, and economics as labels and discarded all documents with multiple labels, which results in a $685\,071$-article data set. We computed tf-idf features on the $2000$ most frequent words to represent all articles. A random subset of $10\,000$ documents is then sampled.
- **STL10:** It contains color images of 96-by-96 pixel size. There are 10 classes with $13\,000$ examples each (Coates et al., 2011). As pre-processing, we extracted features from the STL-10 image data set using a ResNet-50 (He et al., 2016), as in previous works (Jiang et al., 2017).
- **Newborn echo cardiograms:** The data set consists of 305 infant echo cardiogram videos from the Hospital Barmherzige Brüder Regensburg. The videos are taken from several different angles, denoted by [LA, KAKL, KAPAP, KAAP, 4CV]. We cropped the videos by isolating the cone of the echo cardiogram, we resized them to 64x64 pixels and split them into individual frames obtaining a total of $N = 20000$ images. The data used is highly sensitive patient data, hence we only use data with informed consent available. Approval for reuse of the data for our research was obtained from the responsible Ethics Committees. In addition, all data is pseudonymized.
- **UTKFace:** This data set contains over $20\,000$ images of male and female face of individuals from 1 to 118 years old, with multiple ethnicities represented (Zhang et al., 2017).

## B   Conditional ELBO Derivations

### B.1   Proof of Lemma 1

1. *The C-ELBO $\mathcal{L}_{\mathrm{C}}$ is a lower bound of the marginal log-likelihood conditioned on $\boldsymbol{W}$, that is*

$$\log p(\mathbf{X}|\boldsymbol{W}) \geq \mathcal{L}_{\mathrm{C}}(\boldsymbol{\theta}, \boldsymbol{\phi}, \boldsymbol{\nu}, \boldsymbol{\pi}, \mathbf{X}|\boldsymbol{W}).$$

2. $\log p(\mathbf{X}|\boldsymbol{W}) = \mathcal{L}_{\mathrm{C}}(\boldsymbol{\theta}, \boldsymbol{\phi}, \boldsymbol{\nu}, \boldsymbol{\pi}, \mathbf{X}|\boldsymbol{W})$ *if and only if* $q_{\boldsymbol{\phi}}(\mathbf{Z}, \mathbf{c}|\mathbf{X}) = p(\mathbf{Z}, \mathbf{c}|\mathbf{X}, \boldsymbol{W})$.

*Proof.* The marginal log-likelihood conditioned on $\boldsymbol{W}$ can be written as

$$\log p(\mathbf{X}|\boldsymbol{W}) = \mathbb{E}_{q_{\boldsymbol{\phi}}(\mathbf{Z},\mathbf{c}|\mathbf{X})} \log p(\mathbf{X}|\boldsymbol{W}) \tag{13}$$

$$= \mathbb{E}_{q_{\boldsymbol{\phi}}(\mathbf{Z},\mathbf{c}|\mathbf{X})} \left[ \log \frac{p(\mathbf{X}, \mathbf{Z}, \mathbf{c}|\boldsymbol{W})}{p(\mathbf{Z}, \mathbf{c}|\mathbf{X}, \boldsymbol{W})} + q_{\boldsymbol{\phi}}(\mathbf{Z}, \mathbf{c}|\mathbf{X}) - q_{\boldsymbol{\phi}}(\mathbf{Z}, \mathbf{c}|\mathbf{X}) \right] \tag{14}$$

$$= \mathbb{E}_{q_{\boldsymbol{\phi}}(\mathbf{Z},\mathbf{c}|\mathbf{X})} \log \frac{p(\mathbf{X}, \mathbf{Z}, \mathbf{c}|\boldsymbol{W})}{q_{\boldsymbol{\phi}}(\mathbf{Z}, \mathbf{c}|\mathbf{X})} + \mathbb{E}_{q_{\boldsymbol{\phi}}(\mathbf{Z},\mathbf{c}|\mathbf{X})} \log \frac{q_{\boldsymbol{\phi}}(\mathbf{Z}, \mathbf{c}|\mathbf{X})}{p(\mathbf{Z}, \mathbf{c}|\mathbf{X}, \boldsymbol{W})}. \tag{15}$$

The first term corresponds to the C-ELBO. We prove this by using Definition 3 and Eq. 7:

$$\mathbb{E}_{q_{\boldsymbol{\phi}}(\mathbf{Z},\mathbf{c}|\mathbf{X})} \log \frac{p(\mathbf{X}, \mathbf{Z}, \mathbf{c}|\boldsymbol{W})}{q_{\boldsymbol{\phi}}(\mathbf{Z}, \mathbf{c}|\mathbf{X})} = \mathbb{E}_{q_{\boldsymbol{\phi}}(\mathbf{Z},\mathbf{c}|\mathbf{X})} \log \frac{p_{\boldsymbol{\theta}}(\mathbf{X}|\mathbf{Z}) p(\mathbf{Z}, \mathbf{c}|\boldsymbol{W}; \boldsymbol{\nu}, \boldsymbol{\pi})}{q_{\boldsymbol{\phi}}(\mathbf{Z}, \mathbf{c}|\mathbf{X})} \tag{16}$$

$$= \mathbb{E}_{q_{\boldsymbol{\phi}}(\mathbf{Z},\mathbf{c}|\mathbf{X})} \left[ \log p_{\boldsymbol{\theta}}(\mathbf{X}|\mathbf{Z}) \right] + \mathbb{E}_{q_{\boldsymbol{\phi}}(\mathbf{Z},\mathbf{c}|\mathbf{X})} \left[ \log \frac{p(\mathbf{Z}, \mathbf{c}|\boldsymbol{W}; \boldsymbol{\nu}, \boldsymbol{\pi})}{q_{\boldsymbol{\phi}}(\mathbf{Z}, \mathbf{c}|\mathbf{X})} \right] \tag{17}$$

$$= \mathbb{E}_{q_{\boldsymbol{\phi}}(\mathbf{Z}|\mathbf{X})} \left[ \log p_{\boldsymbol{\theta}}(\mathbf{X}|\mathbf{Z}) \right] - D_{KL}(q_{\boldsymbol{\phi}}(\mathbf{Z}, \mathbf{c}|\mathbf{X}) \| p(\mathbf{Z}, \mathbf{c}|\boldsymbol{W}; \boldsymbol{\nu}, \boldsymbol{\pi})) \tag{18}$$

$$= \mathcal{L}_{\mathrm{C}}(\boldsymbol{\theta}, \boldsymbol{\phi}, \boldsymbol{\nu}, \boldsymbol{\pi}, \mathbf{X}|\boldsymbol{W}). \tag{19}$$

Combining Eq. 15 with Eq. 19, the following holds:

$$\log p(\mathbf{X}|\boldsymbol{W}) = \mathcal{L}_{\mathrm{C}}(\boldsymbol{\theta}, \boldsymbol{\phi}, \boldsymbol{\nu}, \boldsymbol{\pi}, \mathbf{X}|\boldsymbol{W}) + D_{KL}(q_{\phi}(\mathbf{Z}, \mathbf{c}|\mathbf{X}) \| p(\mathbf{Z}, \mathbf{c}|\mathbf{X}, \boldsymbol{W})). \tag{20}$$

Given the non-negativity of the Kullback–Leibler divergence, that is $D_{KL}(q\|p) \geq 0$, it follows that $\log p(\mathbf{X}|\boldsymbol{W}) \geq \mathcal{L}_{\mathrm{C}}(\boldsymbol{\theta}, \boldsymbol{\phi}, \boldsymbol{\nu}, \boldsymbol{\pi}, \mathbf{X}|\boldsymbol{W})$. Finally, given that $D_{KL}(q\|p) = 0$ if and only if $q(\cdot) = p(\cdot)$ the second part of the Lemma follows. $\qquad\square$

## B.2 Proof of Lemma 2

*The Conditional ELBO $\mathcal{L}_{\mathrm{C}}$ factorizes as follow:*

$$\mathcal{L}_{\mathrm{C}}(\boldsymbol{\theta}, \boldsymbol{\phi}, \boldsymbol{\nu}, \boldsymbol{\pi}, \mathbf{X}|\boldsymbol{W}) = -\log \Omega(\boldsymbol{\pi}) + \sum_{i=1}^{N} E_{q_{\phi}(\mathbf{z}_i|\mathbf{x}_i)} \Big[ \log p_{\boldsymbol{\theta}}(\mathbf{x}_i|\mathbf{z}_i) - \log q_{\phi}(\mathbf{z}_i|\mathbf{x}_i) \Big]$$

$$+ \sum_{i=1}^{N} E_{q_{\phi}(\mathbf{z}_i|\mathbf{x}_i)} \sum_{k=1}^{K} p(k|\mathbf{z}_i) \Big[ \log p(\mathbf{z}_i|k) + \log \pi_{\mathrm{k}} - \log p(k|\mathbf{z}_i) \Big] \tag{21}$$

$$+ \sum_{i \neq j=1}^{N} E_{q_{\phi}(\mathbf{z}_i|\mathbf{x}_i)} E_{q_{\phi}(\mathbf{z}_j|\mathbf{x}_j)} \sum_{k=1}^{K} p(k|\mathbf{z}_i) p(k|\mathbf{z}_j) \boldsymbol{W}_{i,j},$$

*where $p(k|\mathbf{z}_i) = p(\mathrm{c}_i = k|\mathbf{z}_i; \boldsymbol{\nu}, \boldsymbol{\pi})$ and $p(\mathbf{z}_i|k) = p(\mathbf{z}_i|\mathrm{c}_i = k; \boldsymbol{\nu})$.*

*Proof.* Using Definition 3 and Eq. 7, the Conditional ELBO can be further factorized as:

$$\mathcal{L}_{\mathrm{C}}(\boldsymbol{\theta}, \boldsymbol{\phi}, \boldsymbol{\nu}, \boldsymbol{\pi}, \mathbf{X}|\boldsymbol{W}) = E_{q_{\phi}(\mathbf{Z}|\mathbf{X})}[\log p_{\boldsymbol{\theta}}(\mathbf{X}|\mathbf{Z})] + E_{q_{\phi}(\mathbf{Z}, \mathbf{c}|\mathbf{X})}[\log p(\mathbf{Z}|\mathbf{c}; \boldsymbol{\nu})]$$
$$+ E_{q_{\phi}(\mathbf{Z}, \mathbf{c}|\mathbf{X})}[\log p(\mathbf{c}|\boldsymbol{W}; \boldsymbol{\pi})] - E_{q_{\phi}(\mathbf{Z}, \mathbf{c}|\mathbf{X})}[\log q_{\phi}(\mathbf{Z}, \mathbf{c}|\mathbf{X})]. \tag{22}$$

By plugging in the variational distribution of Eq. 9, the C-ELBO reads:

$$\mathcal{L}_{\mathrm{C}}(\boldsymbol{\theta}, \boldsymbol{\phi}, \boldsymbol{\nu}, \boldsymbol{\pi}, \mathbf{X}|\boldsymbol{W}) = \mathbb{E}_{q_{\phi}(\mathbf{Z}|\mathbf{X})}[\log p_{\boldsymbol{\theta}}(\mathbf{X}|\mathbf{Z})] + \mathbb{E}_{q_{\phi}(\mathbf{Z}|\mathbf{X})p(\mathbf{c}|\mathbf{Z}; \boldsymbol{\nu}, \boldsymbol{\pi}))}[\log p(\mathbf{Z}|\mathbf{c}; \boldsymbol{\nu})]$$
$$+ \mathbb{E}_{q_{\phi}(\mathbf{Z}|\mathbf{X})p(\mathbf{c}|\mathbf{Z}; \boldsymbol{\nu}, \boldsymbol{\pi})}[\log p(\mathbf{c}|\boldsymbol{W}; \boldsymbol{\pi})] - \mathbb{E}_{q_{\phi}(\mathbf{Z}|\mathbf{X})}[\log q_{\phi}(\mathbf{Z}|\mathbf{X})] \tag{23}$$
$$- \mathbb{E}_{q_{\phi}(\mathbf{Z}|\mathbf{X})p(\mathbf{c}|\mathbf{Z}; \boldsymbol{\nu}, \boldsymbol{\pi})}[\log p(\mathbf{c}|\mathbf{Z}; \boldsymbol{\nu}, \boldsymbol{\pi})].$$

The third term depends on $\boldsymbol{W}$ and is investigated in the following. Given that $q_{\phi}(\mathbf{Z}|\mathbf{X})p(\mathbf{c}|\mathbf{Z}) = \prod_i q_{\phi}(\mathbf{z}_i|\mathbf{x}_i)p(\mathrm{c}_i|\mathbf{z}_i)$ and using Definition 2, $E_{q_{\phi}(\mathbf{Z}, \mathbf{c}|\mathbf{X})}[\log p(\mathbf{c}|\boldsymbol{W}]$ can be factorized as

$$E_{q_{\phi}(\mathbf{Z}, \mathbf{c}|\mathbf{X})}[\log p(\mathbf{c}|\boldsymbol{W})] = E_{q_{\phi}(\mathbf{Z}, \mathbf{c}|\mathbf{X})} \log \frac{1}{\Omega(\boldsymbol{\pi})} \prod_i \pi_{\mathrm{c}_i} \prod_{j \neq i} \exp\left(\boldsymbol{W}_{i,j} \delta_{\mathrm{c}_i \mathrm{c}_j}\right) \tag{24}$$

$$= -\log \Omega(\boldsymbol{\pi}) + \sum_{i=1}^{N} E_{q_{\phi}(\mathbf{z}_i, \mathrm{c}_i|\mathbf{x}_i)} \log \pi_{\mathrm{c}_i} + \sum_{i,j=1, i \neq j}^{N} E_{q_{\phi}(\mathbf{z}_i, \mathrm{c}_i|\mathbf{x}_i)} E_{q_{\phi}(\mathbf{z}_j, \mathrm{c}_i|\mathbf{x}_j)} \boldsymbol{W}_{i,j} \delta_{\mathrm{c}_i \mathrm{c}_j}, \tag{25}$$

By observing that $E_{q_{\phi}(\mathbf{z}_i, \mathrm{c}_i|\mathbf{x}_i)}(\cdot) = E_{q_{\phi}(\mathbf{z}_i|\mathbf{x}_i)} \sum_{k=1}^{K} p(\mathrm{c}_i = k|\mathbf{z}_i; \boldsymbol{\nu}, \boldsymbol{\pi})(\cdot)$ the last term of Eq. 25 can be written as:

$$\sum_{i,j=1, i \neq j}^{N} E_{q_{\phi}(\mathbf{z}_i|\mathbf{x}_i)} E_{q_{\phi}(\mathbf{z}_j|\mathbf{x}_j)} \sum_{k=1}^{K} p(\mathrm{c}_i = k|\mathbf{z}_i; \boldsymbol{\nu}, \boldsymbol{\pi}) \sum_{h=1}^{K} p(\mathrm{c}_i = h|\mathbf{z}_j; \boldsymbol{\nu}, \boldsymbol{\pi}) \boldsymbol{W}_{i,j} \delta_{k,h} \tag{26}$$

$$= \sum_{i,j=1, i \neq j}^{N} E_{q_{\phi}(\mathbf{z}_i|\mathbf{x}_i)} E_{q_{\phi}(\mathbf{z}_j|\mathbf{x}_j)} \sum_{k=1}^{K} p(\mathrm{c}_i = k|\mathbf{z}_i; \boldsymbol{\nu}, \boldsymbol{\pi}) p(\mathrm{c}_i = k|\mathbf{z}_j; \boldsymbol{\nu}, \boldsymbol{\pi}) \boldsymbol{W}_{i,j} \tag{27}$$

Given the above equations, Eq. can be further factorized as

$$
\begin{aligned}
\mathcal{L}_{\mathrm{C}}(\mathbf{X}|\boldsymbol{G}) = &\sum_{i=1}^{N} \mathbb{E}_{q_\phi(\mathbf{z}_i|\mathbf{x}_i)}[\log p_\theta(\mathbf{x}_i|\mathbf{z}_i)] \\
&+ \sum_{i=1}^{N} \mathbb{E}_{q_\phi(\mathbf{z}_i|\mathbf{x}_i)}\Big[ \sum_{k=1}^{K} p(\mathbf{c}_i = k|\mathbf{z}_i) \log p(\mathbf{z}_i|\mathbf{c}_i = k) \Big] \\
&- \log \Omega(\boldsymbol{\pi}) + \sum_{i=1}^{N} \mathbb{E}_{q_\phi(\mathbf{z}_i|\mathbf{x}_i)} \sum_{k=1}^{K} p(\mathbf{c}_i = k|\mathbf{z}_i) \log \pi_{\mathbf{c}_i} \\
&+ \sum_{i,j=1,i\neq j}^{N} \mathbb{E}_{q_\phi(\mathbf{z}_i|\mathbf{x}_i)} \mathbb{E}_{q_\phi(\mathbf{z}_j|\mathbf{x}_j)} \sum_{k=1}^{K} p(\mathbf{c}_i = k|\mathbf{z}_i) p(\mathbf{c}_j = k|\mathbf{z}_j) \boldsymbol{W}_{i,j} \\
&- \sum_{i=1}^{N} \mathbb{E}_{q_\phi(\mathbf{z}_i|\mathbf{x}_i)}[\log q(\mathbf{z}_i|\mathbf{x}_i)] \\
&- \sum_{i=1}^{N} \mathbb{E}_{q_\phi(\mathbf{z}_i|\mathbf{x}_i)} \sum_{k=1}^{K} p(\mathbf{c}_i = k|\mathbf{z}_i)[\log p(\mathbf{c}_i = k|\mathbf{z}_i)].
\end{aligned}
\tag{28}
$$

where we simplified $p(\mathbf{c}|\mathbf{z}_i; \boldsymbol{\nu}, \boldsymbol{\pi})$ in $p(\mathbf{c}|\mathbf{z}_i)$ for clarity. By re-ordering the terms the above equation is equal to Eq. 21. □

### B.3 Optimisation with the SGVB estimator

Using the SGVB estimator (Kingma & Welling, 2014; Rezende et al., 2014), we can approximate the C-ELBO defined in Eq. 11 as:

$$
\begin{aligned}
\mathcal{L}_{\mathrm{C}}(\mathbf{X}|\boldsymbol{G}) = &\sum_{i=1}^{N} \frac{1}{L} \sum_{l=1}^{L} \Big[ \log p_\theta(\mathbf{x}_i|\mathbf{z}_i^{(l)}) - \log q_\phi(\mathbf{z}_i^{(l)}|\mathbf{x}_i) \\
&+ \sum_{k=1}^{K} p(\mathbf{c}_i = k|\mathbf{z}_i^{(l)}) \log p(\mathbf{z}_i^{(l)}|\mathbf{c}_i = k) + \sum_{k=1}^{K} p(\mathbf{c}_i = k|\mathbf{z}_i^{(l)}) \log \pi_{\mathbf{c}_i} \\
&- \sum_{k=1}^{K} p(\mathbf{c}_i = k|\mathbf{z}_i^{(l)}) \log p(\mathbf{c}_i = k|\mathbf{z}_i^{(l)}) \Big] \\
&+ \sum_{i,j=1,i\neq j}^{N} \sum_{k=1}^{K} p(\mathbf{c}_i = k|\mathbf{z}_i^{(l)}) p(\mathbf{c}_j = k|\mathbf{z}_j^{(l)}) \boldsymbol{W}_{i,j},
\end{aligned}
\tag{29}
$$

where $\log \Omega(\boldsymbol{\pi})$ is treated as constant and removed from the objective, $L$ is the number of Monte Carlo samples in the SGVB estimator and it is set to $L = 1$ in all experiments.

## C  Further possible Constraints

Given the flexibility of our general framework, different types of constraints can be included in the formulation of the weighting functions $g_i(\mathbf{c})$. In particular, we could include triple-constraints by modifying the weighting function to be:

$$
g_i(\mathbf{c}) = \prod_{j,k\neq i} \exp\left( \boldsymbol{W}_{i,j,k} \delta_{\mathbf{c}_i \mathbf{c}_j \mathbf{c}_k} \right)
\tag{30}
$$
$$
\text{with} \quad \boldsymbol{W} \in \mathbb{R}^{N \times N \times N} \text{ symmetric.}
$$

where $\boldsymbol{W}_{i,j,k} = 0$ if we do not have any prior information, $\boldsymbol{W}_{i,j,k} > 0$ indicates that the samples $\boldsymbol{x}_i$, $\boldsymbol{x}_j$ and $\boldsymbol{x}_k$ should be clustered together and $\boldsymbol{W}_{i,j,k} < 0$ if they should belong to different clusters. The analysis of these different constraints formulation is outside the scope of our work but they may represent interesting directions for future work.

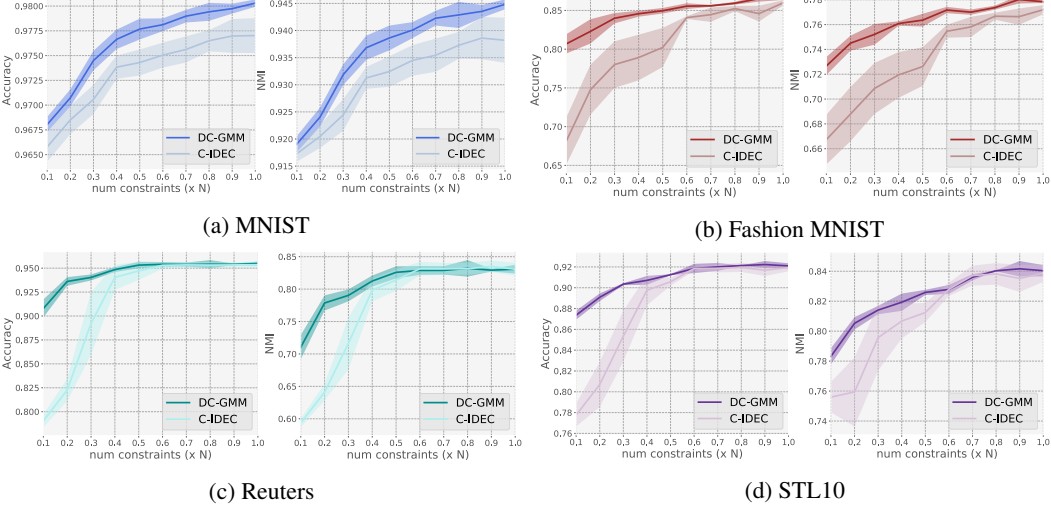



(a) MNIST        (b) Fashion MNIST

(c) Reuters        (d) STL10



Figure 6: Clustering performance on four different data sets with the number of constraints varying between $0.1 \times N$ and $N$, where $N$ is the length of the data set. Accuracy and NMI are used as evaluation metrics.

## D SCDC Comparison

We perform a through comparison with the SCDC baseline. For a fair comparison we modified the provided code (Luo et al., 2018) to include 6000 constraints to match the setting of the rest of the baselines. In Table 3 we report the accuracy, Normalized Mutual Information (NMI), and Adjusted Rand Index (ARI) for MNIST (LeCun et al., 2010) and Fashion MNIST (Xiao et al., 2017). As the results were highly unstable, we reported only the best performance over 10 runs. The code provided by the authors performed very poorly on the Reuters and the STL-10 data sets as it has been optimized for the MNIST dataset, hence we decided to exclude them.

Table 3: Clustering performances of SCDC (Luo et al., 2018) with 6000 pairwise constraints. The maximum value is computed across 10 runs with different random model initialization.

| Dataset | Acc | NMI | ARI |
|---|---|---|---|
| MNIST | 70.9 | 81.0 | 72.4 |
| FASHION | 51.0 | 59.6 | 39.5 |

## E Further Experiments

### E.1 Different number of constraints

We plot the clustering performance in terms of accuracy and Normalized Mutual Information using a varying number of constraints, $N_c$, in Figure 6. We observe that our method outperfoms C-IDEC by a large margin on all four data sets when fewer constraints are used. When $N_c$ gets close to $N$, the two methods tend to saturate on the Reuters and STL data sets.

### E.2 Noisy Labels

In Fig 7, we present the results in term of Accuracy and Normalized Mutual Information of both our model, DC-GMM, and the strongest baseline, C-IDEC with $N$ noisy constraints. In particular, the results are computed for $q \in \{0.1, 0.2, 0.3\}$, where $q$ determines the fraction of pairwise constraints with flipped signs (that is, when a must-link is turned into a cannot-link and vice-versa).

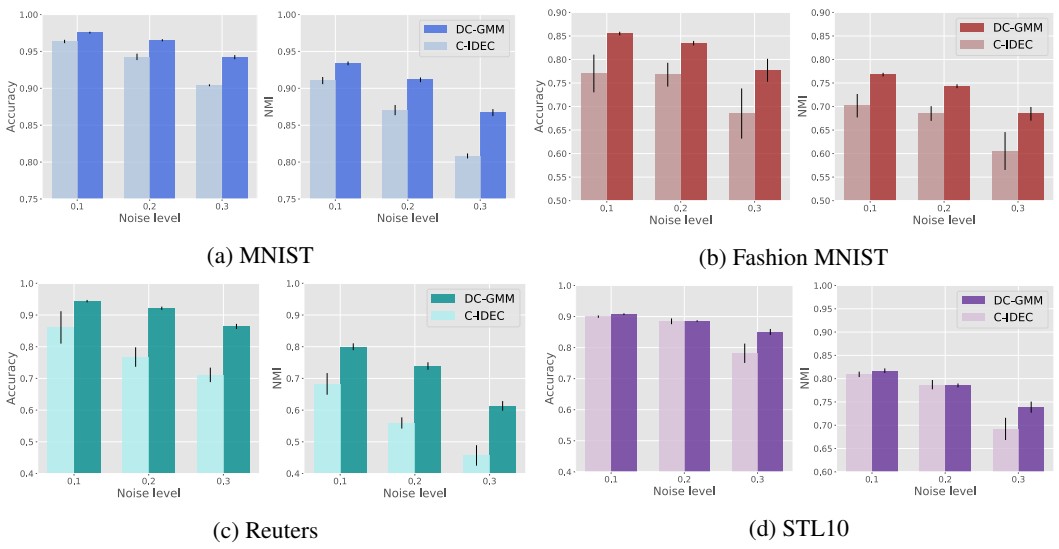

(a) MNIST

(b) Fashion MNIST

(c) Reuters

(d) STL10

Figure 7: Accuracy and NMI clustering performance on four different data sets with noisy labels.

## E.3 Face Image Generation

We evaluate the generative capabilities of our model using the UTKFace data set (Zhang et al., 2017). Using the multivariate Gaussian distributions of each cluster in the learned embedded space, we test the generative capabilities of our method, DC-GMM, by first recovering the mean face of each cluster, and then generating several more faces from each cluster. Figure 8 shows these generated samples. As can be observed, the ethnicities present in the data set are represented well by the mean face. Furthermore, the sampled faces all correspond to the respective cluster, and have a good amount of variation. The quality of generated samples could be improved by using higher resolution training samples or different CNN architectures.

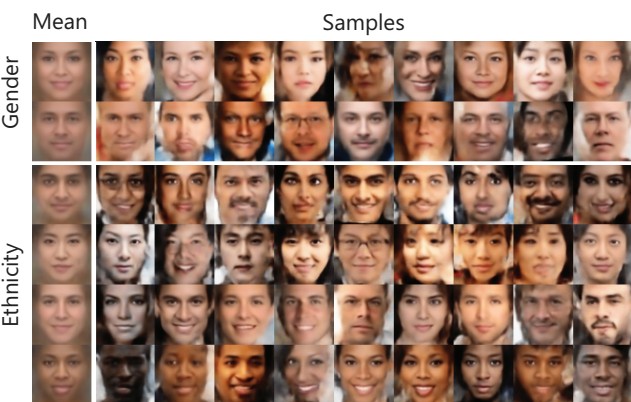

Figure 8: Mean face and sampled faces for each learned cluster, top two rows corresponding to gender, bottom rows to ethnicity

## F Implementation Details

### F.1 Hyper-parameters setting

In Table 4 we specify the hyper-parameters setting of our model, DC-GMM. Given the semi-supervised setting, we did not focus in fine-tuning the hyper-parameters but rather we chose standard

configurations for all data sets. The learning rate is set to 0.001 and it decreases every 20 epochs with a decay rate of 0.9. Additionally, we observed that our model is robust against changes in the hyper-parameters.

Table 4: Hyperparameters setting of our model, DC-GMM.

|                | MNIST | FASHION | REUTERS | STL10 |
|----------------|-------|---------|---------|-------|
| Batch size     | 256   | 256     | 256     | 256   |
| Epochs         | 1000  | 500     | 500     | 500   |
| Learning rate  | 0.001 | 0.001   | 0.001   | 0.001 |
| Decay          | 0.9   | 0.9     | 0.9     | 0.9   |
| Epochs decay   | 20    | 20      | 20      | 20    |

## F.2   Heart Echo

In addition to the model described in Section 4, we also used a VGG-like convolutional neural network. This model is implemented in Tensorflow, using two VGG blocks (using a $3 \times 3$ kernel size) of 32 and 64 filters for the encoder, followed by a single fully-connected layer reducing down to an embedding of dimension 10. The decoder has a symmetric architecture.

The VAE is pretrained for 10 epochs, following which our model is trained for 500 epochs using the same hyper-parameters of Table 4. Refer to the accompanying code for further details.

## F.3   Face Image Generation

The face image generation experiments using the UTK Face data set described in Section 4 were carried out using VGG-like convolutional neural networks implemented in Tensorflow. In particular, the input image size of $64 \times 64 \times 3$ allowed two VGG blocks (using a $3 \times 3$ kernel size) of 64 and 128 filters for the encoder, followed by a single fully-connected layer reducing down to an embedding of dimension 50. The decoder has a symmetric architecture.

The VAE is pretrained for 10 epochs, following which our model is trained for 1000 epochs using a batch size of 256, a learning rate of 0.001 that decreases every 50 epochs with a decay reate of 0.9. Refer to the accompanying code for further details.

## F.4   Resource Usage

Experiments were conducted on an internal computing cluster. Each experiment configuration used one NVIDIA GPU (either a 1080TI or 2080TI), 4 CPUs and a total of 20GB of memory.