# OpenReview forum: "Deep Conditional Gaussian Mixture Model for Constrained Clustering"
_NeurIPS.cc/2021/Conference — NeurIPS 2021 Poster_

### Official Review · Reviewer_yJfq · 2021-07-11

**Rating:** 5
**Confidence:** 4

**Summary:**

This paper provides a new probabilistic constrained clustering algorithm, Deep Conditional Gaussian Mixture Model (DC-GMM).
DC-GMM is based on VaDE or GMM-VAE, but DC-GMM adopts a new prior to incorporate the prior information, constrained clustering. They provide the prior formulation in definition 2, and its corresponding evidence lower bound term in Eq. (11).
DC-GMM shows better performance on MNIST, Fashion, Reuters, STL-10 datasets. Furthermore, they compare their methods and baselines on realistic settings, constrained clustering with noisy labels.

**Ethical Concerns:**

There are no ethical concerns.

**Limitations And Societal Impact:**

The authors adequately addressed the limitations of the paper.

**Main Review:**

< strength >
1. Constrained clustering is an important but not well-explored research area. This paper contributes an important research area, constrained clustering.
2. The proposed methods are so simple, and they can be adapted to many models.
3. They provide realistic experiments, realistic settings (noisy labels), and a realistic dataset (Heart Echo).

< weakness >
1. Table1 shows that the DC-GMM performs better than other methods. However, the intuitive description or reason is not well explained.
2. SCDC is one of the most important baselines, but there is no reported value for Fashion, Reuters, and STL-10 in Table 1
3. This paper focuses on probabilistic clustering to utilized the learned representation to the other tasks. However, this paper only performs clustering tasks.

< question >
1. There can be many candidate ways to construct prior p(c|W;\pi).
What are the intuition and the advantage of prior modeling in Eq. (4) instead of other candidate ways?
2. The value of SCDC in Table1 is taken from the original paper. However, the experiment settings of this paper and the SCDC paper are something different. Is it a fair evaluation?
3. There are other directions to incorporate relation information into the model [1,2,3,4]. I suggest that the authors include the discussion and experimental comparison.
4. There are recent works for constrained clustering (even though the methods do not focus on deep learning).
It will be interesting that DC-GMM performs better than the combination of recent advanced VAE + recent advanced constrained clustering algorithm


< reference >
1. Xu, Hongteng, et al. "Learning autoencoders with relational regularization." International Conference on Machine Learning. PMLR, 2020.
2. Mylonas, Charilaos, Imad Abdallah, and Eleni Chatzi. "Relational VAE: A Continuous Latent Variable Model for Graph Structured Data." arXiv preprint arXiv:2106.16049 (2021).
3. Bai, Liang, JiYe Liang, and Fuyuan Cao. "Semi-supervised clustering with constraints of different types from multiple information sources." IEEE transactions on pattern analysis and machine intelligence (2020).
4. Tang, Da, et al. "Correlated variational auto-encoders." International Conference on Machine Learning. PMLR, 2019.

**Time Spent Reviewing:**

6

---

> ### Author Response · Authors · 2021-08-07
> **Response to Reviewer yJfq**
>
> We thank the reviewer for the detailed feedback. Below we respond to each question raised by the reviewer:
> 1) The main advantages are three-fold: it naturally accommodates both hard constraints and soft preference, it results in a highly interpretable ELBO (see Eq. 11) and it shows remarkable empirical results.
> 2) We spent considerable time tuning the SDCD baseline to match our experiment setting, however the results were remarkably lower than those reported in their paper (see Appendix D for the SDCD results with our setting). Additionally, the authors did not publish the entire code but only a subset of it, which was applicable only to black and white images such as MNIST and fMNIST. For these reasons, we decided to only report the original results, which were obtained with a higher number of constraints than those used in our setting (we checked the original code to count the number of constraints used).
> 3) We thank the reviewer for the additional references, we will gladly incorporate them in the related work and in the experiment comparison (if applicable) in the Camera-Ready version.
> 4) We investigated several combinations of recent clustering methods applied to the learned latent space of VAEs. From the results we obtained, it was often the case that the embedded points did not follow a clear clustering structure if the training was not performed end-to-end (see Fig 4 and 5). The reason is that the VAE by itself focuses only on the overall reconstruction of the original samples. A certain constrained clustering problem, on the other hand, could be driven by aspects of the data that could be easily ignored by the VAE, as they may have a limited contribution to the reconstruction.
>
> Finally, while it is true that we mainly focused on clustering tasks, we also explored the generative capabilities of our model using the UTKFace data set (see Appendix E.3).

---

> > ### Comment · Reviewer_yJfq · 2021-08-18
> > **Response to Author**
> >
> > Thank you for your reply.
> > 3. Could you provide the discussion or comparison between your models and above reference?
> > 4. Figure 4 and Figure 5 support that the necessity of end-to-end method. The results of Figure 4 and Figure 5 are based on the VaDE and PCA, and I don't think their methods are fancy. I suggest that the author provide the results of more advanced methods.

---

> > > ### Author Response · Authors · 2021-08-24
> > > **Response to Reviewer yJfq**
> > >
> > > We thank the reviewer for the additional reply.
> > >
> > > 3. We investigated the provided references as suggested by the reviewer. [1] minimizes the discrepancy between the prior and the posterior distribution of a VAE using the Gromov-Wasserstein distance. Different from our method, it is an unsupervised method and it does not use additional side information. Hence, the comparison is not applicable. [2] and [3], on the other hand, have been published after our submission. By inspecting their results, we could only compare to [3] as one common dataset (MNIST) was used in their experiments. We noticed that their approach has (a) a higher computational complexity 0(n^3) and (b) remarkably lower empirical performance (see Fig. 2-r). Finally, [4] represents a more interesting comparison as the authors proposed to augment the prior distribution of the VAE to take into account the correlations between samples. However, we noticed two substantial differences in the two approaches. The first one lies in the proposed prior, which, contrary to our approach, requires additional lower bounds as the ELBO cannot be directly applied. The second difference lies in both the unobserved variables and the generative process. Different from [4], DC-GMM assumes the existence of the unobserved latent variable, c, also known as cluster assignment, which characterizes the distribution of the latent variable z. On the contrary, [4] directly models the distribution of z. For the above reason, [4] does not focus on clustering and it provides only one clustering experiment on a rather simple synthetic dataset. To measure the model performance on such a setting, they compute the cluster assignments after the model is trained, by performing a PCA over the latent embeddings z and looking at their coefficient ranks.
> > > 4. We thanked the reviewer for the suggestion. We used PCA only to visualize the latent space of the VaDE/ GMM-VAE. Given Fig. 4/5 we believe that any constrained clustering method would fail if based on the euclidean distance metric. Hence the need for a latent variable model. We would appreciate any suggestions/references for the more advanced VAEs / constrained clustering algorithms that we should employ.

---

### Official Review · Reviewer_5L8k · 2021-07-11

**Rating:** 7
**Confidence:** 4

**Summary:**

The paper introduces a new method for constrained clustering with varying degree of certainty. The framework is based on the variational auto-encoder. The numerical experiments are extensive.

**Limitations And Societal Impact:**

Yes

**Main Review:**

Strengths. The paper is well and clearly written. The numerical experiments are well conducted. The technical details are provided.

Weaknesses. Although I think that the contribution is interesting and important, my major concern is the setting of the considered problem. The matrix W that reflects the prior knowledge of the connections between data points and also degree of confidence, is in some sense, the direct indicator on how the points should be clustered. It seems that W turns, in some sense, the problem of clustering into a classification problem, since the information provided by W is very strong and rich. Probably this explains such a good empirical performance?

My next question is very related to my previous concern. If a human expert who provides W, indicated which points are closely related and which are not, what is the principled difference if the expert provides directly class labels? I would appreciate some examples of real applications showing that the matrix W provides significantly different information from supervised learning.


**Time Spent Reviewing:**

4

---

> ### Author Response · Authors · 2021-08-07
> **Response to Reviewer 5L8k**
>
> We thank the reviewer for the feedback. Below we respond to each question raised by the reviewer:
> 1) We would argue that the information provided by W is not as rich as in the classic semi-supervised scenario. The reason lies behind the fact that pairwise constraints are composed of both must-link constraints and cannot-links. The latter only indicates that two data points should not be clustered together. Let’s say we have 10 clusters (as in MNIST) and we have a cannot-link between $x_1$ and $x_2$. If we know for certain that $x_1$ is in cluster 1, then $x_2$ could still be in 9 different clusters. As the dataset is annotated by sampling randomly pairs of samples, 90% of the pairwise constraints will be cannot-links and only 10% must-links. So it follows that the more clusters we have (1) the less information the cannot-link constraints contain and (2) the more frequently we will sample cannot-links rather than must-links.
> 2) Pairwise constraints have a major advantage over labels especially in the crowdsourcing setting, which is a way of annotating data sets from a large pool of people. Due to the lack of expertise of the crowd, it has been shown that comparing samples is easier than directly labeling them. For example, let’s assume the task is to classify many different species of trees. It might be easier to show a pair of samples to non-experts and ask them whether they look similar or not, rather than to ask the specific type. Additionally, it is always possible to automatically convert labels into pairwise annotations, hence the additional information could also come as a combination of labels and pairwise constraints, depending on the level of expertise of the annotators.

---

### Official Review · Reviewer_2qnh · 2021-07-16

**Rating:** 4
**Confidence:** 5

**Summary:**

This paper developed a new generative clustering method. Compared with existing methods, this paper incorporates the pairwise prior knowledge, and then derived ELBO as the objective function to learn model parameters. The authors then conduct experiments to verify the performance of the proposed method.

**Limitations And Societal Impact:**

Not applicable.

**Main Review:**

This paper developed a new generative clustering method. Compared with existing methods, this paper incorporates the pairwise prior knowledge, and then derived ELBO as the objective function to learn model parameters. The authors then conduct experiments to verify the performance of the proposed method.

Pros:
1. This paper is well written and easy to follow.

2. The authors conduct extensive experiments to verify the performance of the proposed paper.

3. The problem that incorporates the prior knowledge is important.

Cons:
1. The novelty is limited. In fact, incorporating the pairwise prior knowledge is trivial here and it is easy to derive the lower bound by following existing works.  From Eq.(11-12), it can be seen that the difference between this method and existing methods is the weight W. If W=I, then it is exactly VAE.

The authors should clearly state the challenge when incorporating the pairwise prior knowledge.

2. The prior knowledge in this paper is fixed. Hence, the contribution is incremental. A more practical choice is to assume that W is a random variable. But this paper didn't study this general scenario and therefore the contribution is limited.

3. This paper missed an important related work [1], which also studied the pairwise prior knowledge.

[1]. Correlated Variational Auto-Encoders

Da Tang, Dawen Liang, Tony Jebara, Nicholas Ruozzi; Proceedings of the 36th International Conference on Machine Learning, PMLR 97:6135-6144

2.


**Time Spent Reviewing:**

3

---

> ### Author Response · Authors · 2021-08-07
> **Response to Reviewer 2qnh**
>
> We thank the reviewer for the feedback. Below we respond to each concern raised by the reviewer:
> 1) We would like to remark that if W=1 then DC-GMM is not equal to the VAE, but rather to the VaDE/GMM-VAE. Additionally, we would strongly argue that the difference between the proposed method and the existing literature does not lie exclusively on the weights W. DC-GMM proposes a novel and well-motivated paradigm to incorporate pairwise constraints within a DGM (see 2.1-3), for which the evidence lower bound is then derived. Following similar reasoning to the one provided by the reviewer, one could also argue that the only difference between the VaDE/GMM-VAE and the VAE lies in the mixing parameter π, as if π is set to a unit vector the two methods are equal.
> 2) We explored configurations with W as a random variable, however, the additional complexity did not result in a more interpretable model or in a better performance.
> 3) We thank the reviewer for the additional reference and we will gladly include it in the Camera-Ready version of the paper. After investigating the provided reference, we noticed two substantial differences in the two approaches. The first one lies in the proposed prior, which, contrary to our approach, requires additional lower bounds as the ELBO cannot be directly applied. The second difference lies in both the unobserved variables and the generative process. Different from [1], DC-GMM assumes the existence of the unobserved latent variable, c, also known as the cluster assignment, which characterizes the distribution of the latent variable z. On the contrary, [1] directly models the distribution of z (see Eq. 4). For the above reason, [1] does not focus on clustering and it provides only one clustering experiment on a rather simple synthetic dataset. To measure the model performance on such a setting, they compute the cluster assignments after the model is trained, by performing a PCA over the latent embeddings z and looking at their coefficient ranks.

---

### Official Review · Reviewer_2A4z · 2021-07-17

**Rating:** 7
**Confidence:** 4

**Summary:**

The paper proposes a method for *constrained clustering* (a semi-supervised setting in which pairs of samples are labeled as belonging to the same or to different clusters). The method is based on learning a GMM prior for the latent space of a VAE where the cluster assignments depend on the pairwise constrains. The authors develop a conditional version of the ELBO for the constrained case and show how it can be optimized using SGVB and reparameterization, like in regular VAE. The proposed method (DC-GMM) is shown to achieve best clustering results for six different datasets (e.g. MNIST where 6000 random training pairs where used as pairwise constraints) compared to recent *constrained clustering* methods and to be more robust to label noise and to small amount of supervision.

**Limitations And Societal Impact:**

The authors discuss the main limitations of their method. I'm not aware of any potential negative social impact.

**Main Review:**

**Originality:** As the authors indicate, the idea of learning a GMM as the prior distribution of the VAE latent space is not by itself new (an additional example of deep clustering with a GMM prior is [1] below). The setting of a GMM latent prior conditional on the pairwise constraints matrix and the proposed optimization method seems to be novel and interesting.

**Clarity:** The paper is very well written - the motivation is well explained, the setting is clearly described as a graphical model and the method is clearly and accurately explained.

**Quality:** The proposed method seems to be technically sound. The experiments are thoroughly conducted and support the claims well. Keeping the number of constrains fixed at 6000 seemed a bit strange to me, but the authors have one result (figure 2) showing the affect of varying the number of constraints.

**Significance:** Overall, the paper seems to be an important contribution to the constrained clustering research.

The fact the that mixture probability distribution $\pi$ is fixed at $1/K$ limits, in my opinion, the usability of the proposed method. The datasets used in the experiments section all have more or less equal cluster probabilities, but in real cases there might be a strong imbalance.

Although, as the authors describe, the size of the supervision is only a small fraction of all possible pairs, it is still significant both practically (manually labeling 6000 pairs) and theoretically - for example, if the pairwise constraints define chains of samples that must reside in the same clusters (e.g. $(x_1, x_2), (x_2, x_3), ...$), then $N$ constrains are sufficient to completely define the clusters. A large number of pairwise constrains, however, seems to be the typical case in the *constrained clustering* literature.

Minor comments:
- It took me a while to understand why the cluster assignments random variable $c$ in the PGM in figure 1 is not part of the plate (I think because individual assignments are not iid). A short explanation can help.


[1] Multi-Modal Deep Clustering: Unsupervised Partitioning of Images, Shiran and Weinshall 2020


**Time Spent Reviewing:**

6

---

> ### Author Response · Authors · 2021-08-07
> **Response to Reviewer 2A4z**
>
> We thank the reviewer for the detailed feedback. We will add a short explanation on Fig. 1 as suggested correctly by the reviewer. We would also like to highlight that we computed the transitive closure over all must-linked points for each experiment. We noticed that the number of additional must-links derived from the given set was often negligible. The reason is that the pairwise annotations are randomly selected by sampling two data points and assigning a must-link if they belong to the same cluster or a cannot-link otherwise. Given that the number of clusters is always bigger or equal than 4 in all experiments, there is always a higher number of cannot-links, which contain less information. Finally, the mixing parameter π can in practice be set to any value (and not just to the uniform distribution), according to the prior information about the dataset.

---

> > ### Comment · Reviewer_2A4z · 2021-08-25
> > **Response to Authors**
> >
> > I want to thank the authors for the additional clarifications and maintain my original "accept" recommendation.

---

### Decision · Program_Chairs · 2021-09-28

**Decision:**

Accept (Poster)

**Comment:**

Reviews have many positive comments about the proposed approach to incorporating pairwise constraints into prior work on VAE-based clustering, which allows for fairly simple inference but nevertheless has good empirical performance.  There are some concerns about significance of the results, but agreement that technical details are novel.  Some reviewers raised questions about conceptual comparisons to related work, that were reasonably addressed by the author response; please be sure to include these comparisons in future revisions.

**Consistency Experiment:**

NeurIPS has a long history of experimentation. In 2014, NeurIPS ran an experiment in which 10% of submissions were reviewed by two independent committees to quantify the randomness in the review process. This year, we repeated a variant of this experiment to see how the quality of the review process has changed over time.  This paper was part of the experiment and was therefore assigned to two committees (consisting of reviewers, an Area Chair, and a Senior Area Chair) that reached independent decisions.  If both committees made the same recommendation, this recommendation was followed. If a single committee recommended acceptance, the paper was accepted (with the exception of a few cases in which the other committee identified what we considered a fatal flaw, e.g., an error in a key result).

This copy’s committee reached the following decision: **Accept (Poster)**

The other committee assigned to the paper recommended **Reject**.  You can find the other set of reviews, along with any follow up discussion with the authors here:
https://openreview.net/forum?id=Blq2djlaP9U